# Prediction of Cardiac Transthyretin Amyloidosis: Electrocardiographic Parameters and the Ratio of Posterior Wall Thickness to the Minimum QRS Complex Voltage in Limb Leads

**DOI:** 10.3390/biomedicines13102493

**Published:** 2025-10-13

**Authors:** Monika Gawor-Prokopczyk, Marta Lipowska, Agnieszka Sioma, Anton Chrustowicz, Jan Henzel, Jacek Grzybowski, Justyna Szczygieł, Anna Wójcik, Marek Konka, Ewa Kowalik, Anna Teresińska, Łukasz Mazurkiewicz

**Affiliations:** 1Department of Cardiomyopathy, National Institute of Cardiology, Alpejska 42, 04-628 Warsaw, Poland; 2Department of Neurology, Medical University of Warsaw, Banacha 1a, 02-097 Warsaw, Poland; 3Department of Coronary and Structural Heart Diseases, National Institute of Cardiology, Alpejska 42, 04-628 Warsaw, Poland; 4Department of Congenital Heart Diseases, National Institute of Cardiology, Alpejska 42, 04-628 Warsaw, Poland; 5Department of Nuclear Medicine, National Institute of Cardiology, Alpejska 42, 04-628 Warsaw, Poland

**Keywords:** cardiac imaging, echocardiography, electrocardiography, cardiac amyloidosis, score, T-amylo, TCAS, transthyretin amyloidosis

## Abstract

**Background/Objectives**: Several predictive models have been proposed to estimate the probability of cardiac transthyretin amyloidosis (ATTR-CA). The aim of our study was to evaluate the usefulness of electrocardiographic parameters, as well as parameters consisting of a combination of myocardial thickness and QRS voltage, as potential predictors of ATTR-CA. **Methods**: In 2018–2025, 285 consecutive patients with suspected cardiac amyloidosis were evaluated, including blood tests, standard 12-lead electrocardiography, transthoracic echocardiography, and [^99m^Tc]Tc-DPD scintigraphy. **Results**: The ratio of posterior wall thickness to minimum QRS voltage in limb leads (PWT/minQRS ratio) as well as several ECG-derived parameters were independent predictors of ATTR-CA. In a comparison of ROC curves, PWT/minQRS ratio exceeded both the minimum and maximum voltage of QRS complexes in limb leads, demonstrated similar predictive value to TCAS and T-amylo scores, and had similar or superior predictive characteristics to posterior wall thickness. A cut-off of >3.3 for PWT/minQRS ratio was as accurate as the published cut-offs for TCAS score ≥6, T-amylo score ≥7, and posterior wall thickness ≥14 mm. In the subgroup of patients with myocardial thickness of at least 15 mm, PWT/minQRS ratio >3.3 was superior to posterior wall thickness ≥14 mm and T-amylo score ≥7 and had similar predictive value for ATTR-CA as TCAS score ≥6. **Conclusions**: In a cohort of undifferentiated patients referred for [^99m^Tc]Tc-DPD scintigraphy due to suspected cardiac amyloidosis, PWT/minQRS ratio showed strong predictive value for ATTR-CA, which was even more pronounced in the subgroup of patients with increased myocardial thickness.

## 1. Introduction

Cardiac transthyretin amyloidosis (ATTR-CA) is a life-threatening, progressive disease [1,2]. The diagnostic algorithm using bone scintigraphy with ^99m^Technetium-3,3-diphosphono-1,2-propanodicarboxylic acid ([^99m^Tc]Tc-DPD) in the absence of monoclonal gammopathy greatly facilitated the diagnosis of patients with ATTR-CA [1,2]. The European Society of Cardiology (ESC) Working Group on Myocardial and Pericardial Diseases proposed that the diagnostic evaluation of cardiac amyloidosis (CA) should be performed in patients with a myocardial thickness of at least 12 mm in the presence of at least one “red flag” or in special populations, such as patients with heart failure with preserved ejection fraction or with aortic stenosis, especially in those over 65 years of age [2,3,4]. Due to the large population of patients meeting these criteria, guideline-compliant management requires numerous examinations including [^99m^Tc]Tc-DPD scintigraphy, prolonging the waiting time for patients with ATTR-CA to receive diagnostic evaluation and, more importantly, specific treatment. To improve diagnosis and maintain a high yield of [^99m^Tc]Tc-DPD scintigraphy, several predictive models have been proposed in recent years to estimate the probability of ATTR-CA prior to [^99m^Tc]Tc-DPD scintigraphy and more precisely select patients for further diagnosis [5,6,7,8]. Some models are based solely on echocardiographic parameters, including non-routine ones, while others take into account only basic clinical and echocardiographic data and are called “simple” by their authors. The increased wall thickness (IWT) score is entirely derived from echocardiographic parameters, and consists of relative wall thickness (RWT), E/e’ ratio, longitudinal strain (LS), tricuspid annulus planar systolic excursion (TAPSE), and four-chamber septal apical to base ratio (SAB) [5]. The Mayo Clinic transthyretin cardiac amyloidosis score (TCAS) comprises age, sex, RWT, posterior wall thickness, left ventricular ejection fraction (LVEF), and history of hypertension [6]. The so-called T-amylo score consists of age, sex, presence of carpal tunnel syndrome, hypertrophy, and low QRS voltage [7]. The published cut-offs considered high risk for ATTR-CA are IWT ≥ 8, TCAS ≥ 6, and T-amylo ≥ 7. The IWT and TCAS scores have been validated in an external study, which showed that posterior wall thickness performed as well or better than more complex multiparametric predictive models mentioned above [8]. Increased posterior wall thickness with a cut-off of ≥14 mm turned out to be a robust and simple predictor of ATTR-CA [8]. Based on this information and given that decreased QRS voltage to LV mass ratio is one of the “red flags” suggestive of CA, it would be useful to validate parameters related to posterior wall thickness in combination with QRS complex voltage to predict ATTR-CA [2,3,4,8]. Although there are reports focusing on electrocardiographic features in ATTR-CA patients, data on the possible use of ECG parameters other than low QRS voltage or pseudonecrosis pattern in the ATTR-CA diagnosis are limited [9,10,11].

The aim of our study was to evaluate, in patients with suspected CA from our center, the usefulness of electrocardiographic parameters, such as minimum and maximum voltage of QRS complexes in limb leads, as well as parameters consisting of a combination of myocardial thickness and QRS voltage, as potential predictors of ATTR-CA, and to compare these parameters with TCAS, T-amylo scores, and increased posterior wall thickness with cut-off of ≥14 mm, three predictive models for ATTR-CA that include only basic and routinely available data.

## 2. Materials and Methods

Consecutive patients with suspected CA who underwent [^99m^Tc]Tc-DPD bone scintigraphy between August 2018 and January 2025 at the National Institute of Cardiology, Warsaw, Poland were evaluated retrospectively. The suspicion of CA was raised in patients with left ventricular (LV) wall thickness of at least 12 mm and at least 1 “red flag” or clinical scenario, as described in detail previously [2,3,4]. In addition to [^99m^Tc]Tc-DPD bone scintigraphy, all patients underwent a routine assessment consisting of medical history, physical examination, blood tests, standard 12-lead electrocardiography (ECG), and transthoracic echocardiography. Blood tests included N-terminal pro-B-type natriuretic peptide (NT-proBNP) and high-sensitivity cardiac troponin T (hs-cTnT) measurements, as well as free light chain testing and immunofixation of serum and urine to exclude light chain amyloidosis (AL). The *TTR* gene sequencing was offered to all patients with ATTR-CA, as described previously [12]. ATTR-CA was diagnosed using a non-invasive algorithm, which confirms the condition when grade 2 or 3 myocardial radiotracer uptake is observed on [^99m^Tc]Tc-DPD bone scintigraphy and no monoclonal protein is found in the serum or urine [1]. Patients with abnormal results of free light chain testing or immunofixation of serum and urine underwent hematologic evaluation, including tissue biopsy (bone marrow, labial salivary gland, gastric, surgical fat tissue, or cardiac biopsy) and immunohistochemical typing of amyloid.

Minimum and maximum voltages of QRS complexes in limb leads were calculated. Standard definitions were used for the interpretation of 12-lead ECGs. Low QRS voltage was defined as QRS amplitude <0.5 mV in all limb leads or <1 mV amplitude in all precordial leads. A pseudonecrosis pattern in precordial leads was defined as pathological Q waves (1/4 R amplitude) or QS waves on 2 consecutive leads in the absence of known ischemic heart disease [9,10]. Patients with ventricular stimulation on ECG were excluded from evaluation of QRS complex voltage amplitude or pseudonecrosis pattern. Similarly, patients with LBBB were excluded from evaluation of pseudonecrosis pattern. The TCAS and T-amylo scores were calculated [6,7]. The RWT was defined as sum of septal and posterior wall thickness divided by left ventricular end diastolic diameter [6]. In addition, the ratio of posterior wall thickness to the minimum voltage of QRS complexes in limb leads (PWT/minQRS ratio) was also calculated. All data were collected retrospectively from medical records and analyzed using an anonymous dataset.

Statistical analyses were performed using MedCalc 12.1.4.0 software (MedCalc, Mariakerke, Belgium). After checking for normal distribution with the Shapiro–Wilk test, continuous variables were compared using either the one-way analysis of variance (ANOVA) or the Kruskal–Wallis test, as appropriate. The normally distributed continuous variables are shown as mean (SD) and non-parametrically distributed variables are shown as median (interquartile range [IQR]). Categorical variables are expressed as frequency (percentage) of patients and were compared using the χ2 test or the Fisher exact test. A 2-sided *p*-value below 0.05 was considered significant. Univariable logistic regression analysis was performed to identify potential predictors of the presence of ATTR-CA. Selected variables with a *p*-value of less than 0.10 in the univariate analysis were included in the multivariate regression model using the backward selection method. Model calibration was assessed by performing the Hosmer–Lemeshow goodness-of-fit test. Receiver operating characteristic (ROC) curve analysis and the area under curve (AUC) were used to evaluate the performance for identifying subjects with ATTR-CA. DeLong’s test was used to compare the areas under two or more correlated ROC curves.

## 3. Results

Of the 285 patients with suspected cardiac amyloidosis, 64 (22.4%) were diagnosed with ATTR-CA, including 15 patients with hereditary ATTR-CA and 36 (12.6%) patients with cardiac AL amyloidosis (AL-CA group), and 185 (64.9%) patients had no evidence of CA (excluded CA group) (Figure 1). A diagnosis of AL amyloidosis was confirmed by immunohistochemical typing of amyloid in the biopsied tissue. In addition to cardiac involvement, AL patients showed evidence of gastrointestinal tract involvement, as demonstrated by positive immunohistochemical findings in labial salivary gland or gastric biopsies. Due to abnormal results of free light chain testing or immunofixation of serum and urine, 15 patients with ATTR-CA and 18 patients from the excluded CA group underwent hematologic evaluation, including tissue biopsy.

The clinical data of the three groups are shown in Table 1. There were more men in the ATTR-CA group, and patients were older, had lower BMI, and more frequently suffered from carpal tunnel syndrome and coronary artery disease than those in the other two groups. Arterial hypertension was present more frequently in the group with excluded CA. The cardiac biomarkers NT-proBNP and hs-cTnT were higher in the groups of patients with CA. On echocardiography, the maximum wall thickness and posterior wall thickness were also higher in the patients with CA.

Five patients with ATTR-CA and 10 patients with excluded CA whose ECG showed ventricular stimulation were excluded from the evaluation of QRS complex voltage amplitude or pseudonecrosis pattern. ATTR-CA patients more frequently had ECG findings characteristic of cardiac amyloidosis, such as pseudonecrosis pattern in precordial leads or low QRS voltage. Minimum and maximum voltages of QRS complexes in limb leads were lower on ECG of the patients with CA. Features of LV hypertrophy on ECG were present more frequently in the group with excluded CA. Patients with ATTR-CA had higher T-amylo and TCAS scores than patients in the other two groups. All patients with CA more frequently had posterior wall thickness of at least 14 mm than patients with excluded CA. The distribution plot of the PWT/minQRS ratio in ATTR-CA patients versus those without ATTR-CA (including patients with excluded CA and those with AL-CA) is presented in Figure 2.

ATTR-CA patients were stratified by the presence of carpal tunnel syndrome (Table 2).

To identify independent determinants of the presence of ATTR-CA, logistic regression was performed on patients with ATTR-CA and patients with excluded CA. Several ECG-derived parameters, such as low QRS voltage, pseudonecrosis pattern in precordial leads, nonspecific intraventricular conduction delay (NIVCD), and right bundle branch block (RBBB), as well as minimum and maximum voltage of QRS complexes in limb leads, were independent predictors of ATTR-CA in univariable analysis (Table 3). In addition, PWT/minQRS ratio was also an independent predictor of ATTR-CA in univariable analysis (Table 3).

Comparative analysis of PWT/minQRS ratio, minimum and maximum voltage of QRS complexes in limb leads was performed (Table 4). In a pairwise comparison of ROC curves, PWT/minQRS ratio exceeded both the minimum and maximum voltage of QRS complexes in limb leads (*p* < 0.001) (Table 4).

Multivariate analysis with the backward selection method was undertaken using parameters included in the TCAS and T-amylo scores, as well as parameters derived from the ECG, which were independent predictors of ATTR-CA in univariate analysis (Table 5). The predictive model consisted of posterior wall thickness, minimum voltage of QRS complexes in limb leads, age, sex, presence of carpal tunnel syndrome, pseudonecrosis pattern, and RBBB, and had an AUC of 0.97 (95% CI: 0.94–0.99). The model showed good calibration as assessed by the Hosmer–Lemeshow goodness-of-fit test (χ2 10.9, *p* = 0.2).

In multivariate analysis using PWT/minQRS ratio and parameters included in the TCAS and T-amylo scores, the independent predictors of ATTR were PWT/minQRS ratio, age, sex, RWT, LVEF, presence of carpal tunnel syndrome, and absence of arterial hypertension (Table 4). The model had AUC of 0.94 (95% CI: 0.9–0.97) and also presented good calibration as assessed by the Hosmer–Lemeshow goodness-of-fit test (χ2 13.3, *p* = 0.1).

Comparative analysis of PWT/minQRS ratio with TCAS, T-amylo scores, and posterior wall thickness was performed using continuous values. There was no significant difference between PWT/minQRS ratio and TCAS and T-amylo scores (*p* = 0.32 and *p* = 0.99 respectively) (Table 6). Posterior wall thickness was outperformed by PWT/minQRS ratio (*p* = 0.008) (Table 6).

The optimal cut-off of PWT/minQRS ratio was >3.3 with a sensitivity of 86.4% and a specificity of 71.3% (AUC 0.87, 95% CI: 0.82–0.91, *p* < 0.001) (Table 7).

A cut-off of >3.3 for PWT/minQRS ratio was as accurate as the published cut-offs for TCAS score ≥6, T-amylo score ≥7, and posterior wall thickness ≥14 mm (*p* = 0.51 and *p* = 0.11, *p* = 0.16 respectively) (Table 8) (Figure 3). All four models had similar predictive value for ATTR-CA (Table 8).

In the subgroup of patients with myocardial thickness of at least 15 mm (53 patients with ATTR-CA and 112 patients with excluded CA), PWT/minQRS ratio >3.3 (AUC 0.82, 95% CI: 0.76–0.88) was superior to posterior wall thickness ≥14 mm (AUC 0.7, 95% CI: 0.63–0.77, *p* = 0.002) and T-amylo score ≥7 (AUC 0.73, 95% CI: 0.66–0.79, *p* = 0.03), and had similar predictive value for ATTR-CA as TCAS score ≥6 (AUC 0.77, 95% CI: 0.7–0.83, *p* = 0.07) (Table 9) (Figure 4).

In the final step, logistic regression was performed on patients with ATTR-CA versus the group without ATTR-CA (including patients with excluded CA and those with AL-CA) (Table 10).

Multivariate analysis with the backward selection method was undertaken using parameters included in the TCAS and T-amylo scores, as well as parameters derived from the ECG, which were independent predictors of ATTR-CA in univariate analysis (Table 11). The predictive model consisted of low QRS voltage, minimum voltage of QRS complexes in limb leads, age, sex, presence of carpal tunnel syndrome, pseudonecrosis pattern, and RBBB, and had an AUC of 0.94 (95% CI: 0.89–0.96). The model showed good calibration as assessed by the Hosmer–Lemeshow goodness-of-fit test (χ2 6.3, *p* = 0.6). In multivariate analysis using PWT/minQRS ratio and parameters included in the TCAS and T-amylo scores, the independent predictors of ATTR were PWT/minQRS ratio, age, sex, LVEF, presence of carpal tunnel syndrome, and low QRS voltage (Table 11). The model had AUC of 0.88 (95% CI: 0.84–0.92) but presented week calibration as assessed by the Hosmer–Lemeshow goodness-of-fit test (χ2 21.7, *p* = 0.005).

Comparative analysis of PWT/minQRS ratio, and minimum and maximum voltages of QRS complexes in limb leads, was performed on patients with ATTR-CA versus the group without ATTR-CA (Table 12). In a pairwise comparison of ROC curves, PWT/minQRS ratio exceeded both the minimum and maximum voltage of QRS complexes in limb leads (*p* < 0.001 and *p* = 0.01, respectively) (Table 12).

Comparative analysis of PWT/minQRS ratio with TCAS, T-amylo scores, and posterior wall thickness was performed using continuous values and cut-offs (Table 13). There was no significant difference between PWT/minQRS ratio and TCAS and T-amylo scores (Table 13). Posterior wall thickness was outperformed by PWT/minQRS ratio in the comparison of continuous values but not in the comparison of cut-offs (Table 13).

## 4. Discussion

Thanks to recent advances in the diagnosis of ATTR-CA and greater awareness of the disease, ATTR-CA is now being diagnosed in an increasing number of patients and, importantly, more often in the early stages of the disease. In addition, new and effective therapies have emerged to further improve the prognosis and quality of life of ATTR-CA patients [2,3,4]. In view of the fact that ATTR-CA treatment is more effective in the early stages of the disease, promptness of diagnosis is crucial [13]. The search for predictive models and novel diagnostic tools or biomarkers for use prior to [^99m^Tc]Tc-DPD scintigraphy may further improve the management of patients with ATTR-CA and shorten the time to diagnosis and appropriate treatment [5,6,7,8,14,15,16]. The multiparametric scores for ATTR-CA prediction are helpful, especially when all the required parameters are close at hand. Notably, simple measurements of some single parameters performed as well as more complex models [8].

Our study, which analyzes usefulness of electrocardiographic derived parameters, such as minimum and maximum voltages of QRS complexes in limb leads, as well as parameters consisting of a combination of myocardial thickness and QRS voltage as potential predictors of ATTR-CA, follows this pragmatic path to broaden the spectrum of easily available diagnostic tools for ATTR-CA.

Both ECG and echocardiography remain essential diagnostic procedures to support a suspicion of CA and initiate further diagnostics [2,3,4]. While echocardiographic features have been widely studied to predict ATTR-CA, ECG seems to have received less attention, although it is one of the most accessible, rapid, and non-invasive tests [5,8,9,10,17,18,19,20]. Low QRS voltage and pseudonecrosis are well-known ‘red flags” of CA, and their predictive characteristics have been determined [9,10,18,19,20]. In previous reports, low QRS voltage was more common in AL amyloidosis compared with ATTR-CA [9,19,20]. In addition, no differences in the incidence of pseudonecrosis pattern among subtypes of amyloidosis were detected [9,19,20]. Moreover, ATTR-CA patients showed a higher prevalence of low QRS voltage and pseudonecrosis pattern in comparison to patients with severe aortic stenosis, hypertrophic cardiomyopathy, and hypertensive heart disease [18]. In our study, pseudonecrosis pattern and low QRS voltage were more frequent among patients with ATTR-CA compared to patients with excluded CA but also compared to patients with AL-CA amyloidosis. This finding may result from the fact that some patients diagnosed with AL-CA amyloidosis at our center, especially in more advanced stages, were directly referred for hematologic treatment without [^99m^Tc]Tc-DPD scintigraphy. Our work resembles the previous findings that patients with ATTR-CA more often show intraventricular conduction abnormalities [9]. Several ECG-derived parameters, among them NIVCD and RBBB, were independent predictors of ATTR-CA in univariable analysis. In addition, RBBB, along with minimum voltage of QRS complexes in limb leads and pseudonecrosis pattern, were included in the multiparametric models.

While conducting our study, we observed that although many patients with ATTR-CA do not meet the criterion for low QRS complex voltage, their ECGs are often characterized by reduced QRS complex voltage. Indeed, the minimum and maximum ORS voltages in the limb leads on the ECG of patients with ATTR-CA and AL-CA amyloidosis were significantly lower than those of patients with excluded CA.

In addition, decreased QRS complex voltage on the ECG of patients with CA is often associated with increased myocardial thickness, which can be expressed as the ratio of posterior wall thickness to minimum QRS complex voltage in limb leads. The newly proposed parameter, PWT/minQRS ratio, as well as minimum and maximum voltages of QRS complexes in limb leads, were independent predictors of ATTR-CA in the logistic regression analysis. In a pairwise comparison of ROC curves, PWT/minQRS ratio exceeded both the minimum and maximum voltage of QRS complexes in limb leads, demonstrated similar predictive value to TCAS and T-amylo scores, and had superior predictive characteristics to posterior wall thickness. A cut-off of >3.3 for PWT/minQRS ratio was as reliable as the reported cut-offs for TCAS score ≥6, T-amylo score ≥7, and posterior wall thickness ≥14 mm.

In the subgroup of patients with myocardial thickness of at least 15 mm, PWT/minQRS ratio >3.3 was superior to posterior wall thickness ≥14 mm and T-amylo score ≥7, and had similar predictive value for ATTR-CA as TCAS score ≥6.

In view that a decreased ratio of QRS voltage to LV mass is one of the “red flags” suggestive of CA, the results we obtained may not be surprising. PWT/minQRS ratio has emerged as a noteworthy feature of ATTR-CA, which is worth taking into account in daily practice, especially in patients with increased myocardial thickness. Importantly, PWT/minQRS ratio is easy to calculate, simple to remember, and derived from routinely performed diagnostics and commonly available clinical variables. PWT/minQRS ratio may act as a refined screening tool—functioning as a “super red flag” within the diagnostic algorithm for ATTR-CA. Its simplicity and objectivity make it a promising candidate for early identification of patients warranting further evaluation, especially in settings where resource optimization is critical. Given its entirely imaging- and ECG-based nature, it could be considered a hallmark indicator that is particularly useful in streamlining patient selection for [^99m^Tc]Tc-DPD scintigraphy. By identifying individuals with a higher pre-test probability of ATTR-CA, this metric has the potential to enhance diagnostic efficiency and reduce delays in initiating disease-specific treatment. However, further validation of this parameter is needed, especially in the context of the stage of the disease.

Our study also validated two clinical diagnostic scoring systems for assessing the risk of ATTR-CA, TCAS and T-amylo scores, in both a cohort of undifferentiated patients and a cohort of patients with increased myocardial thickness. Both models demonstrated comparable predictive accuracy for ATTR-CA.

The main limitations of this study are its retrospective and single-center design, which may introduce potential selection bias. Similar to the studies dedicated to the development of the T-amylo and TCAS scores, as well as the posterior wall thickness criterion with a cut-off of ≥14 mm—all of which were also retrospective—we included consecutive patients who underwent both bone scintigraphy and evaluation for clonal dyscrasia. This approach may limit the generalizability of results [6,7,8]. The selection of patients who underwent bone scintigraphy clearly led to underrepresentation of the AL-CA group, as some patients diagnosed with AL-CA amyloidosis—particularly those in more advanced stages—were directly referred for hematologic treatment without undergoing [^99m^Tc]Tc-DPD scintigraphy. The previous studies on development of TCAS score and the posterior wall thickness criterion with a cut-off of ≥14 mm addressed this issue by excluding all AL-CA patients [6,8]. A different strategy was adopted in the T-amylo study, where patients with AL-CA were included in the “ATTR-CA excluded” group; however, only 5 out of 119 patients in that group had AL-CA [7]. None of the studies validated their scoring systems in the AL-CA group [6,7,8]. Our study also lacks proper validation in patients with AL-CA amyloidosis. Although including the AL-CA group made our study more reflective of real-life clinical practice, the relatively small sample size of this subgroup may limit the possibility of making reliable comparisons between study groups. Another limitation is lack of validation of our results in an external, independent prospective cohort. In our study, we compared PWT/minQRS ratio with previously published scores. However, we fully acknowledge the methodological differences between these models, which were derived from distinct types of variables—ranging from imaging and ECG parameters to clinical history. Our aim was to present their relative effectiveness within our cohort. We did not intend to suggest a direct metric-to-metric equivalence, but rather to explore how each performs within a shared diagnostic context.

## 5. Conclusions

This study may facilitate the diagnosis of patients with ATTR-CA. In a cohort of undifferentiated patients referred for [^99m^Tc]Tc-DPD scintigraphy due to suspected CA, PWT/minQRS ratio, a simple and routinely available parameter, showed strong predictive value for ATTR-CA, which was even more pronounced in the subgroup of patients with increased myocardial thickness. The accuracy of PWT/minQRS ratio for ATTR-CA was similar to or better than that of previously proposed predictive models. PWT/minQRS ratio may serve as a sensitive predictor of ATTR-CA.

## Figures and Tables

**Figure 1 biomedicines-13-02493-f001:**
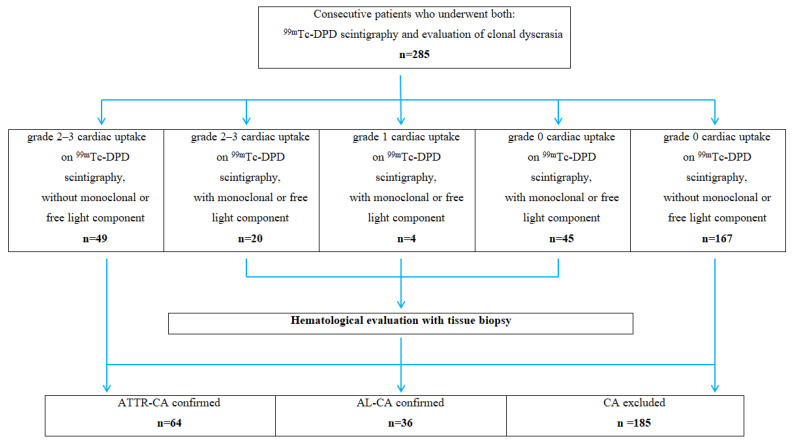
Patient flowchart.

**Figure 2 biomedicines-13-02493-f002:**
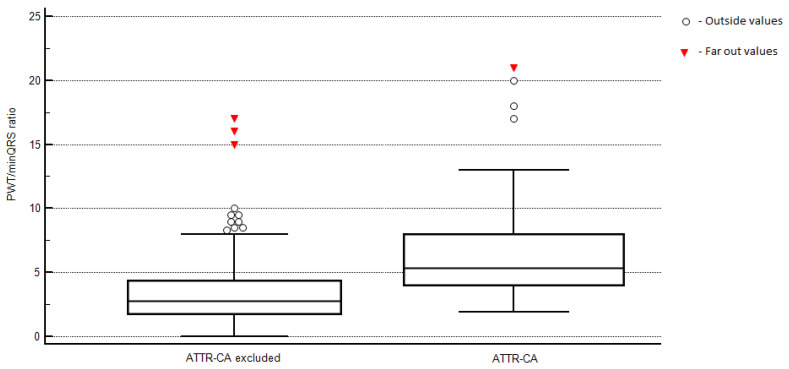
The distribution plot of the PWT/minQRS ratio in ATTR-CA patients versus those without ATTR-CA.

**Figure 3 biomedicines-13-02493-f003:**
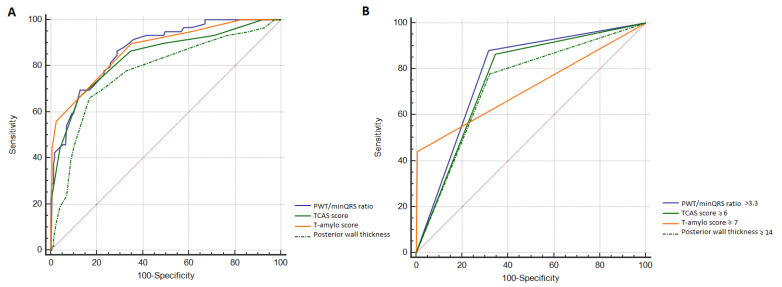
Comparison of receiver operating characteristic curves to diagnose cardiac transthyretin amyloidosis with continuous variables (**A**) and specific cut-offs (**B**) in the ATTR-CA vs. excluded CA group.

**Figure 4 biomedicines-13-02493-f004:**
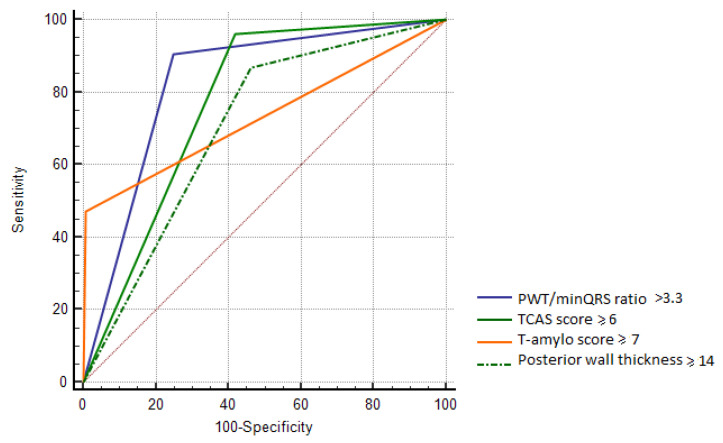
Comparison of receiver operating characteristic curves to diagnose cardiac transthyretin amyloidosis with specific cut-offs in patients with myocardial thickness ≥15 mm.

**Table 1 biomedicines-13-02493-t001:** Baseline clinical characteristics of the patients.

	ATTR-CA (n = 64)	AL-CA (n = 36)	Excluded CA (n = 185)	*p*-Value
Age at diagnosis, y	74 (63.5–79)	66 (56.5–70.5)	67 (55–73)	<0.001
Male, (%)	57 (89)	24 (66.7)	108 (58.4)	<0.001
BMI, kg/m^2^	24 (23–27)	26.9 (23–29.5)	27.6 (24.9–31)	<0.001
NYHA functional class, n (%)				<0.001
I	10 (15.6)	3 (8.3)	32 (17.3)	
II	38 (59.4)	15 (41.7)	127 (68.6)	
III	14 (21.9)	18 (50)	26 (14)	
IV	2 (3.1)	0 (0)	0 (0)	
Coronary artery disease, n (%)	39 (60.9)	11 (30.6)	71 (38.4)	<0.001
Arterial hypertension, n (%)	31 (48.4)	7 (19.4)	136 (73.5)	<0.001
Diabetes mellitus, n (%)	15 (23.4)	5 (13.9)	56 (30.4)	0.09
History of stroke, n (%)	7 (10.9)	4 (11.1)	14 (7.6)	0.62
Carpal tunnel syndrome, n (%)	27 (42.2)	3 (8.3)	9 (4.9)	<0.001
ECG				
Low QRS voltage, n (%) *	18 (30.5)	3 (8.3)	2 (1.1)	<0.001
Minimum ORS voltage in the limb leads, mm *	3 (2–4)	3 (2–4)	5 (3–8)	<0.001
Maximum ORS voltage in the limb leads, mm *	8 (5–9)	7 (5–11)	13 (8–16)	<0.001
AF, n (%)	37 (57.8)	8 (22.2)	91 (49.2)	0.002
AVB (of any degree), n (%)	26 (40.6)	11 (30.5)	56 (30.3)	0.3
LBBB, n (%) *	10 (16.9)	4 (11.1)	13 (7.4)	0.1
RBBB, n (%) *	14 (23.7)	4 (11.1)	16 (9.1)	0.01
NIVCD, n (%) *	10 (16.9)	3 (8.3)	12 (6.8)	0.07
LVH on ECG, n (%) *	7 (11.8)	3 (8.3)	69 (39.4)	0.002
Pseudonecrosis pattern, n (%) *	37 (62.7)	15 (42.8)	16 (9.1)	<0.001
ICD, n %	4 (6.2)	1 (2.8)	39 (21.1)	<0.001
Pacemaker or ICD, n %	13 (20.3)	0 (0)	17 (9.2)	0.004
Echocardiography				
MWT, mm	19 (17–22)	18.5 (16–20.5)	16 (13.7–20)	<0.001
PWT, mm	16 (14–18.5)	16.0 (15–18)	12 (11–14)	<0.001
LA dimension, mm	46 (42–51)	45.5 (42–49)	47 (42.5–53)	0.2
LVEDd, mm	45 (40.5–49.5)	44 (39.2–47.7)	48 (43–54)	<0.001
RWT	0.8 (0.6–1)	0.8 (0.7–1)	0.6 (0.5–0.7)	<0.001
E/e’ ratio	15 (11–18)	18 (13.5–21.5)	12.5 (9–16.5)	<0.001
LVEF, %	50 (45–60)	59.5 (50–65)	60 (53.7–65)	<0.001
TAPSE, mm	17.4 (4.5)	17 (4.4)	20.6 (4.8)	<0.001
hs-cTnT [ng/L]	51.5 (37–77)	57.5 (42–117)	23 (13–40.7)	<0.001
NT-proBNP [pg/mL]	2629 (1193.7–4109)	4912 (1408–7888.5)	1108.5 (324–2743.5)	<0.001
PWT ≥14 mm, %	51 (79.7)	30 (83.3)	59 (31)	<0.001
TCAS score	7 (6–9)	6 (4.5–8)	4 (3–6)	<0.001
T-amylo score *	6 (5–8)	5 (2–5)	3 (2–5)	<0.001
PWT/minORS ratio *	5.3 (4–8)	5.4 (4.3–8.5)	2.4 (1.7–3.7)	<0.001

Data are presented as mean (SD) or median (interquartile range) unless indicated otherwise. * Five patients with ATTR-CA and 10 patients with excluded CA whose ECG showed ventricular stimulation were excluded from the evaluation of QRS complex voltage amplitude, LBBB, RBBB, NIVCD, LVH, or pseudonecrosis pattern.

**Table 2 biomedicines-13-02493-t002:** ATTR-CA patients stratified by the presence of carpal tunnel syndrome.

	ATTR-CA with CTS (n = 24)	ATTR-CA Without CTS (n = 35)	*p*-Value
TCAS score	7 (6–9)	7 (6–8.2)	0.98
T-amylo score	8 (8–9)	5 (5–6)	<0.001
PWT	16.3 (3.6)	16.3 (3.7)	0.98
PWT/minORS ratio	5.1 (3.5–9)	6 (4.5–7.7)	0.72

**Table 3 biomedicines-13-02493-t003:** Predictors of cardiac transthyretin amyloidosis in univariate analysis.

	Coefficient	OR	95% CI	*p*-Value
Age at diagnosis	0.04	1.05	1–1.1	<0.001
Male	1.8	5.8	2.5–13.4	<0.001
Arterial hypertension	−1.1	0.3	0.2–0.6	<0.001
Carpal tunnel syndrome	2.7	14.3	6.2–32.8	<0.001
ECG				
Low QRS voltage	3.6	38	8.6–170.2	<0.001
Minimum ORS voltage in the limb leads	−0.6	0.5	0.4–0.7	<0.001
Maximum ORS voltage in the limb leads	−0.3	0.8	0.7–0.8	<0.001
AF	0.3	1.4	0.8–2.5	0.23
AVB (of any degree)	0.4	1.6	0.9–2.8	0.14
LBBB	0.9	2.4	1–5.7	0.06
RBBB	1.1	3.1	1.4–6.8	0.004
NIVCD	0.9	2.6	1–6.3	0.04
Pseudonecrosis pattern	2.7	15	7.2–31.5	<0.001
Echocardiography				
MWT	0.1	1.1	1–1.2	0.002
PWT	0.3	1.3	1.2–1.5	<0.001
LA dimension	−0.03	1	0.9–1	0.08
LVEDd	−0.1	0.9	0.9–1	0.002
RWT	2.4	11.5	3.5–38	<0.001
E/e’ ratio	0.05	1.05	1–1.1	0.03
LVEF	−0.04	0.9	0.9–1	<0.001
TAPSE	−0.1	0.9	0.8–0.9	<0.001
hs-cTnT	0.01	1	1–1.1	<0.001
NT-proBNP	0.01	1	1–1.1	0.01
PWT/minORS ratio	0.8	2.2	1.7–2.8	<0.001

**Table 4 biomedicines-13-02493-t004:** Comparison of receiver operating characteristic curves for the PWT/minQRS ratio, and minimum and maximum voltages of QRS complexes in limb leads, to diagnose cardiac transthyretin amyloidosis in the ATTR-CA vs. excluded CA group.

	Difference Between Areas	Standard Error	95% CI	*p*-Value
PWT/QRS ratio vs. minimum QRS voltage	0.08	0.02	0.05–0.12	<0.001
PWT/QRS ratio vs. maximum QRS voltage	0.1	0.03	0.05–0.15	<0.001
minimum QRS voltage vs. maximum QRS voltage	0.01	0.02	−0.04–0.06	0.64

**Table 5 biomedicines-13-02493-t005:** Predictors of cardiac transthyretin amyloidosis in multivariate analysis.

	Coefficient	OR	95% CI	*p*-Value
TCAS and T-amylo scores parameters and ECG derived parameters				
PWT	0.34	1.4	1–1.7	0.001
Minimum ORS voltage in the limb leads	−0.93	0.4	0.2–0.6	<0.001
Age	0.06	1.1	1–1.2	0.03
Male	1.8	0.2	0.1–0.9	0.04
Carpal tunnel syndrome	3.1	23.4	4.7–117	<0.001
Pseudonecrosis pattern	3.9	50.7	9.7–265	<0.001
RBBB	2.6	13.1	1.9–87.2	0.008
TCAS and T-amylo scores parameters and PWT/minQRS ratio				
PWT/minORS ratio	0.7	2.1	1.5–32.8	<0.001
Age	0.07	1.1	1–1.1	0.002
Male	1.5	0.2	0.1–0.8	0.02
Carpal tunnel syndrome	3.2	26	6.7–100.4	<0.001
RWT	4.3	72	8.2–643	<0.001
LVEF	−0.06	0.9	0.9–1	0.01
Arterial hypertension	−1.1	3.1	1.1–9.3	0.04

**Table 6 biomedicines-13-02493-t006:** Comparison of receiver operating characteristic curves for the PWT/minQRS ratio, TCAS score, T-amylo score, and posterior wall thickness to diagnose cardiac transthyretin amyloidosis in the ATTR-CA vs. excluded CA group with continuous variables.

	Difference Between Areas	Standard Error	95% CI	*p*-Value
PWT/minQRS ratio vs. TCAS score	0.03	0.03	−0.03–0.1	0.32
PWT/minQRS ratio vs. T-amylo score	0.0005	0.03	−0.06–0.07	0.99
PWT/minQRS ratio vs. posterior wall thickness	0.1	0.04	0.02–0.17	0.008

**Table 7 biomedicines-13-02493-t007:** Receiver operator curve characteristics and cut-off points of parameters for diagnosis of cardiac transthyretin amyloidosis in the ATTR-CA vs. excluded CA group.

	Cut Point	Sensitivity	Specificity	AUC	95%CI	*p*-Value
PWT/minORS ratio	>3.3	86.4	71.3	0.87	0.82–0.91	<0.001
Minimum ORS voltage in the limb leads, mm	<5	88.1	56.3	0.79	0.73–0.84	<0.001
Maximum ORS voltage in the limb leads, mm	<11	84.7	60.3	0.78	0.72–0.83	<0.001
PWT, mm	>15	67.2	83.7	0.79	0.73–0.84	<0.001
TCAS score	>6	68.7	87.6	0.85	0.8–0.89	<0.001
T-amylo score	>5	56.2	97.3	0.87	0.82–0.91	<0.001
PWT ≥14 mm, %	≥1	79.7	68.1	0.74	0.68–0.79	<0.001
TCAS score ≥6	≥1	87.5	64.9	0.76	0.7–0.81	<0.001
T-amylo score ≥7	≥1	45.3	98.9	0.72	0.66–0.78	<0.001

**Table 8 biomedicines-13-02493-t008:** Comparison of receiver operating characteristic curves for the PWT/minQRS ratio, TCAS score, T-amylo score, and posterior wall thickness to diagnose cardiac transthyretin amyloidosis in the ATTR-CA vs. excluded CA group with specific cut-offs.

	Difference Between Areas	Standard Error	95% CI	*p*-Value
PWT/minQRS ratio > 3.3 vs. TCAS score ≥6	0.02	0.03	−0.04–0.09	0.51
PWT/minQRS ratio > 3.3 vs. T-amylo score ≥7	0.06	0.04	−0.01–0.14	0.11
PWT/minQRS ratio > 3.3 vs. posterior wall thickness ≥14 mm	0.05	0.04	−0.02–0.13	0.16

**Table 9 biomedicines-13-02493-t009:** Comparison of receiver operating characteristic curves for the PWT/minQRS ratio, TCAS score, T-amylo score, and posterior wall thickness to diagnose cardiac transthyretin amyloidosis in the subgroup of patients with myocardial thickness of at least 15 mm.

	Difference Between Areas	Standard Error	95% CI	*p*-Value
PWT/minQRS ratio >3.3 vs. TCAS score ≥6	0.05	0.03	−0.01–0.12	0.07
PWT/minQRS ratio >3.3 vs. T-amylo score ≥7	0.09	0.04	0.01–0.18	0.03
PWT/minQRS ratio >3.3 vs. posterior wall thickness ≥14 mm	0.12	0.04	0.05–0.2	0.002

**Table 10 biomedicines-13-02493-t010:** Predictors of cardiac transthyretin amyloidosis in patients with ATTR-CA versus the group without ATTR-CA in univariate analysis.

	Coefficient	OR	95% CI	*p*-Value
Age at diagnosis	0.05	1.05	1–1.1	<0.001
Male	1.7	5.5	2.4–12.6	<0.001
Arterial hypertension	−0.7	0.3	0.3–0.9	0.02
Carpal tunnel syndrome	2.5	12.7	5.9–27.3	<0.001
ECG				
Low QRS voltage	2.8	18	6.3–51.5	<0.001
Minimum ORS voltage in the limb leads	−0.5	0.6	0.5–0.8	<0.001
Maximum ORS voltage in the limb leads	−0.2	0.8	0.7–0.9	<0.001
AF	0.5	1.7	1–3	0.07
AVB (of any degree)	0.4	1.6	0.9–2.8	0.12
LBBB	0.8	2.2	0.9–5.1	0.07
RBBB	1.1	3	1.4–6.3	0.004
NIVCD	0.9	2.5	1–5.8	0.04
Pseudonecrosis pattern	2.2	8.9	4.6–17.5	<0.001
Echocardiography				
MWT	0.1	1.1	1–1.1	0.004
PWT	0.2	1.2	1.2–1.4	<0.001
LA dimension	−0.03	1	0.9–1	0.12
LVEDd	−0.1	0.9	0.9–1	0.009
RWT	1.9	6.5	2.2–19.7	<0.001
E/e’ ratio	0.03	1	1–1.1	0.22
LVEF	−0.03	1	0.9–1	0.001
TAPSE	−0.1	0.9	0.8–1	<0.001
hs-cTnT	0.01	1	1–1.1	0.02
NT-proBNP	0.01	1	1–1.1	0.35
PWT/ minORS ratio	0.3	1.37	1.2–1.5	<0.001

**Table 11 biomedicines-13-02493-t011:** Predictors of cardiac transthyretin amyloidosis in patients with ATTR-CA versus the group without ATTR-CA in multivariate analysis.

	Coefficient	OR	95% CI	*p*-Value
TCAS and T-amylo scores parameters and ECG derived parameters				
Low QRS voltage	1.4	4	0.8–19.6	0.08
Minimum ORS voltage in the limb leads	−0.29	0.75	0.6–1	0.03
Age	0.04	1	1–1.1	0.04
Male	1.5	4.3	1.2–15.2	0.02
Carpal tunnel syndrome	2.9	17.8	4.9–63.8	<0.001
Pseudonecrosis pattern	2.6	14.2	5.1–39.5	<0.001
RBBB	2.5	12.3	2.7–54.5	0.001
TCAS and T-amylo scores parameters and PWT/minQRS ratio				
PWT/minORS ratio	0.2	1.2	1.1–1.35	<0.001
Age	0.04	1.04	1–1.1	0.02
Male	1.3	3.7	1.3–10.6	0.01
Carpal tunnel syndrome	2.7	15	5.5–40.7	<0.001
Low QRS voltage	2	7.5	2.2–25.3	0.001
LVEF	−0.02	1	0.9–1	0.07

**Table 12 biomedicines-13-02493-t012:** Comparison of receiver operating characteristic curves for the PWT/minQRS ratio, and minimum and maximum voltages of QRS complexes in limb leads, to diagnose cardiac transthyretin amyloidosis in the ATTR-CA group vs. the group without ATTR-CA.

	Difference Between Areas	Standard Error	95% CI	*p*-Value
PWT/QRS ratio vs. minimum QRS voltage	0.06	0.02	0.03–0.1	<0.001
PWT/QRS ratio vs. maximum QRS voltage	0.07	0.03	0.02–0.13	0.01
minimum QRS voltage vs. maximum QRS voltage	0.007	0.03	−0.04–0.06	0.8

**Table 13 biomedicines-13-02493-t013:** Comparison of receiver operating characteristic curves for the PWT/minQRS ratio, TCAS score, T-amylo score, and posterior wall thickness to diagnose cardiac transthyretin amyloidosis in the ATTR-CA group vs. the group without ATTR-CA.

	Difference Between Areas	Standard Error	95% CI	*p*-Value
PWT/minQRS ratio vs. TCAS score	0.005	0.03	−0.06–0.07	0.87
PWT/minQRS ratio vs. T-amylo score	0.05	0.04	−0.02–0.12	0.16
PWT/minQRS ratio vs. posterior wall thickness	0.07	0.04	0.05–0.15	0.04
PWT/minQRS ratio > 3.3 vs. TCAS score ≥6	0.006	0.03	−0.06–0.07	0.85
PWT/minQRS ratio > 3.3 vs. T-amylo score ≥7	0.03	0.04	−0.05–0.1	0.49
PWT/minQRS ratio > 3.3 vs. posterior wall thickness ≥14 mm	0.05	0.04	−0.02–0.12	0.14

## Data Availability

All data underlying the results are available within the article.

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
