# Peer review of "Prediction of Cardiac Transthyretin Amyloidosis: Electrocardiographic Parameters and the Ratio of Posterior Wall Thickness to the Minimum QRS Complex Voltage in Limb Leads"

_biomedicines, 2025, doi:10.3390/biomedicines13102493_

Round 1

Reviewer 1 Report

Comments and Suggestions for Authors

The present article shows the predictive value of a simple echocardiographic/ECG parameter (PWT/minQRS) for the diagnosis of ATTR-CA. The paper is interesting, but some aspects require further revisions. These are my comments:

Methods:
- Please specify whether bone scintigraphy was performed in all patients or only in those without a monoclonal gammopathy.
- It is not specified how the diagnosis was made in patients who requested a biopsy-proven diagnosis. Please specify the type of biopsy and how the amyloid typing was performed in the methods.

Results
- Please, report in how may patients a biopsy-proven diagnosis was necessary
- Report whether patients with AL amyloidosis had evidence of organ involvement other than cardiac involvement.
- Table 1: it is not clear how p value was calculated and which groups of patients were compared. Furthermore, I suggest to rename the AL group as AL-CA

- The relatively small sample size may hamper the comparison between the 3 groups. This could be particularly relevant when it comes to compare patients with ATTR and AL amyloidosis. Patients with AL amyloidosis often experience weight loss and are cachectic. Therefore, it is surprising that in this study, BMI was found to be lower in patients with ATTR (considering also the small number of patients with ATTRv). Similarly, differences in ECG patterns between ATTR and AL amyloidosis may be influenced by the small sample size. I recommend to recognize this limitation in the discussion

- I do not fully agree with the design of the logistic regression analysis. Since the aim is the developement of a predictive score for ATTR, I think is not correct to exclude patients with AL amyloidosis from the analysis. I would recommend repeating the logistic regression analysis including patients with AL amyloidosis.

- Figures 2, 3 and 4 are too small to read

Discussion:

The study limitations are not described

Author Response

Thank you very much for taking the time to review this manuscript. Please find the detailed responses below and the corresponding corrections highlighted in the re-submitted files. "Methods:
- Please specify whether bone scintigraphy was performed in all patients or only in those without a monoclonal gammopathy."

Our study group, similar to the T-amylo, TCAS, and posterior wall thickness  with a cut-off of ≥14 mm studies, included consecutive patients who concurrently underwent both bone scintigraphy and evaluation for clonal dyscrasia using free light chain testing and immunofixation of serum and urine.

"- It is not specified how the diagnosis was made in patients who requested a biopsy-proven diagnosis. Please specify the type of biopsy and how the amyloid typing was performed in the methods."

The flow diagram has been revised to present the diagnostic pathways in detail. The following sentences have been added to the Methods section: “Patients with abnormal results of free light chain testing or immunofixation of serum and urine underwent hematologic evaluation, including tissue biopsy (bone marrow, labial salivary gland, gastric, surgical fat tissue, or cardiac biopsy) and immunohistochemical typing of amyloid.”

 "Results
- Please, report in how may patients a biopsy-proven diagnosis was necessary"

The following sentences have been added to the Results section: “Due to abnormal results of free light chain testing or immunofixation of serum and urine, 15 patients with ATTR-CA and 18 patients from “excluded CA” group  underwent hematologic evaluation, including tissue biopsy. “

"- Report whether patients with AL amyloidosis had evidence of organ involvement other than cardiac involvement."

The following sentences have been added to the Results section: “A diagnosis of AL amyloidosis was confirmed by immunohistochemical typing of amyloid in the biopsied tissue. In addition to cardiac involvement, AL patients showed evidence of gastrointestinal tract involvement, as demonstrated by positive immunohistochemical findings in labial salivary gland or gastric biopsies.”

"- Table 1: it is not clear how p value was calculated and which groups of patients were compared."

In Table 1, three groups were compared using either the ANOVA test (if all three groups had a normal distribution) or the Kruskal-Wallis test (if at least one group did not have a normal distribution). Both are statistical methods used to compare the means of three or more groups. The chi-square test was used to compare NYHA class across the three groups. The statistical analysis is described in the Methods section.

"Furthermore, I suggest to rename the AL group as AL-CA"

The AL group has been renamed AL-CA.

"- The relatively small sample size may hamper the comparison between the 3 groups. This could be particularly relevant when it comes to compare patients with ATTR and AL amyloidosis"

I definitely agree. The TCAS study and the posterior wall thickness study with a cut-off of ≥14 mm addressed this issue by excluding AL-CA patients. In the posterior wall thickness ≥14 mm study, 56 AL-CA patients were excluded. In the TCAS study, the number of excluded AL-CA patients was not even mentioned.

A different strategy was adopted in the T-amylo study, where five patients with AL-CA were included in the “ATTR-CA excluded” group, which consisted of 119 patients in total. None of the studies validated their scoring systems (TCAS, T-amylo, or posterior wall thickness ≥14 mm) in the AL-CA group.

From one perspective, we could also exclude the AL-CA group from our study, aligning our methodology with that of the TCAS and posterior wall thickness ≥14 mm studies. I would have no objection to this approach, as it would simplify both the methodology and the message of the study.

"Patients with AL amyloidosis often experience weight loss and are cachectic. Therefore, it is surprising that in this study, BMI was found to be lower in patients with ATTR (considering also the small number of patients with ATTRv). Similarly, differences in ECG patterns between ATTR and AL amyloidosis may be influenced by the small sample size. I recommend to recognize this limitation in the discussion"

Thank you for your comment. The study limitations have been incorporated into the Discussion section.

"- I do not fully agree with the design of the logistic regression analysis. Since the aim is the developement of a predictive score for ATTR, I think is not correct to exclude patients with AL amyloidosis from the analysis. I would recommend repeating the logistic regression analysis including patients with AL amyloidosis."

The methodology of our study was designed similarly to the T-amylo, TCAS, and posterior wall thickness with a cut-off of ≥14 mm studies. As previously mentioned, both the TCAS study and the posterior wall thickness study excluded AL-CA patients. In contrast, the T-amylo study included five patients with AL-CA in the “ATTR-CA excluded” group, which consisted of 119 patients in total. None of these studies validated their scoring systems (TCAS, T-amylo, or posterior wall thickness ≥14 mm) in the AL-CA group. In accordance with your request, I repeated the logistic regression by merging AL-CA and excluded CA patients into the ATTR-CA excluded group.

"- Figures 2, 3 and 4 are too small to read"

The figures have been enlarged.

"Discussion: The study limitations are not described"

The study limitations have been incorporated into the Discussion:

“The main limitations of the study are its retrospective and single-center design, which may introduce potential selection bias. Similar to the studies dedicated to the development of the T-amylo and TCAS scores, as well as the posterior wall thickness criterion with a cut-off of ≥14 mm—all of which were also retrospective—we included consecutive patients who underwent both bone scintigraphy and evaluation for clonal dyscrasia. This approach may limit the generalizability of results [6-8]. Selection of patients who underwent bone scintigraphy clearly led to underrepresentation of the AL-CA group, as some patients diagnosed with AL-CA amyloidosis—particularly those in more advanced stages—were directly referred for hematologic treatment without undergoing [99mTc]Tc-DPD scintigraphy. The previous studies on development of TCAS score and the posterior wall thickness criterion with a cut-off of ≥14 mm addressed this issue by excluding all AL-CA patients [6,8]. A different strategy was adopted in the T-amylo study, where five patients with AL-CA were included in the “ATTR-CA excluded” group. However, none of the studies validated their scoring systems in the AL-CA group [6-8]. Our study also lacks proper validation in patients with AL-CA amyloidosis. Although including the AL-CA group made our study more reflective of real-life clinical practice, the relatively small sample size of this subgroup may limit the possibility of making reliable comparisons between study groups. Another limitation is lack of validation of our results in an external, independent prospective cohort. In our study, we compared the PWT/minQRS ratio with previously published scores. We fully acknowledge the methodological differences between these models, which were derived from distinct types of variables—ranging from imaging and ECG parameters to clinical history. Our aim was to present their relative effectiveness within our cohort. We did not intend to suggest a direct metric-to-metric equivalence, but rather to explore how each performs within a shared diagnostic context."

  1.  

Reviewer 2 Report

Comments and Suggestions for Authors

This is a timely and well-executed study that explores a pragmatic approach to risk stratification for cardiac transthyretin amyloidosis (ATTR-CA) using simple, accessible parameters. The authors propose a novel yet intuitive metric—the ratio of posterior wall thickness to minimum QRS voltage in limb leads (PWT/minQRS)—and demonstrate its incremental diagnostic utility over established predictive scores (TCAS, T-amylo) and standalone parameters. Major comments:

- ATTR was defined non-invasively (DPD grade 2–3 without monoclonal protein), AL by hematologic work-up, and “excluded CA” by absence of scintigraphic/hematologic criteria. Given potential differential verification and incorporation bias, clarify whether any biopsies were performed in indeterminate cases and how equivocal DPD (grade 1) or borderline monoclonal results were handled. Provide a flow diagram with all verification pathways and missingness. 

- ATTRv vs ATTRwt and AL comparator: Stratify performance by genotype (ATTRv vs wt) and by presence of neuropathy/carpal tunnel, which also features in T-amylo. Likewise, because low voltage is more prevalent in AL in some series, clarify whether including AL in model building could distort discrimination between ATTR and “no CA.” A three-class analysis (ATTR vs AL vs no CA) or pairwise one-vs-rest ROC could help.

- The authors compare the PWT/minQRS ratio with TCAS and T-amylo scores. However, the time point, method, and clinical context of score derivation are not fully harmonized across these models. For instance, T-amylo includes carpal tunnel syndrome—an anamnestically derived variable—while PWT/minQRS is entirely imaging/ECG based. It is thus not a pure “metric-to-metric” comparison.

- ROC curves should include 95% CI bands; DeLong comparisons should present ΔAUC with CIs, not only p-values. Please add calibration curves, distribution plots of the ratio by class (ATTR vs non-ATTR), and a Bland–Altman/repeatability plot for voltage measurements if feasible. A transparent model card (TRIPOD adherence table) is recommended. 

- The study is retrospective and single-center, yet these limitations are not sufficiently acknowledged or discussed. The patient selection is based on referral for [99mTc]Tc-DPD scintigraphy, which may lead to referral and spectrum bias. The proportion of patients diagnosed with AL amyloidosis (12.6%) and ATTR-CA (22.4%) raises the question of how representative this cohort is of broader clinical practice. A paragraph in the Discussion should explicitly address the issue of generalizability, particularly how this approach would perform in primary cardiology clinics or in asymptomatic carriers of TTR mutations.

Author Response

Thank you very much for taking the time to review this manuscript. Please find the detailed responses below and the corresponding corrections highlighted in the re-submitted files.

"- ATTR was defined non-invasively (DPD grade 2–3 without monoclonal protein), AL by hematologic work-up, and “excluded CA” by absence of scintigraphic/hematologic criteria. Given potential differential verification and incorporation bias, clarify whether any biopsies were performed in indeterminate cases and how equivocal DPD (grade 1) or borderline monoclonal results were handled. Provide a flow diagram with all verification pathways and missingness."

Thank you for your comment.  The flow diagram has been revised to present the diagnostic pathways in detail. The following sentences have been added to the Methods section: “Patients with abnormal results of free light chain testing or immunofixation of serum and urine underwent hematologic evaluation, including tissue biopsy (bone marrow, labial salivary gland, gastric, surgical fat tissue, or cardiac biopsy) and immunohistochemical typing of amyloid.” The following sentences have been added to the Results section: “A diagnosis of AL amyloidosis was confirmed by immunohistochemical typing of amyloid in the biopsied tissue (…). Due to abnormal results of free light chain testing or immunofixation of serum and urine, 15 patients with ATTR-CA and 18 patients from “excluded CA” group underwent hematologic evaluation, including tissue biopsy. “

"- ATTRv vs ATTRwt and AL comparator: Stratify performance by genotype (ATTRv vs wt) and by presence of neuropathy/carpal tunnel, which also features in T-amylo"

Patients with ATTR were stratified based on the presence of carpal tunnel syndrome (Table 2). We did not perform stratification by genotype, consistent with previous studies (TCAS, T-amylo, posterior wall thickness with a cut-off of ≥14 mm), in which the genetic status of the study population was not considered.

"-Likewise, because low voltage is more prevalent in AL in some series, clarify whether including AL in model building could distort discrimination between ATTR and “no CA.” A three-class analysis (ATTR vs AL vs no CA) or pairwise one-vs-rest ROC could help."

In accordance with your request, we repeated the analysis by merging AL-CA and excluded CA patients into the ATTR-CA excluded group (Table 10-13). 

"- The authors compare the PWT/minQRS ratio with TCAS and T-amylo scores. However, the time point, method, and clinical context of score derivation are not fully harmonized across these models. For instance, T-amylo includes carpal tunnel syndrome—an anamnestically derived variable—while PWT/minQRS is entirely imaging/ECG based. It is thus not a pure “metric-to-metric” comparison."

We appreciate this insightful observation and fully acknowledge the methodological differences between the PWT/minQRS ratio and the previously published scores. We have clarified this point further in the revised Discussion section: “In our study, we compared the PWT/minQRS ratio with previously published scores. We fully acknowledge the methodological differences between these models, which were derived from distinct types of variables—ranging from imaging and ECG parameters to clinical history. Our aim was to present their relative effectiveness within our cohort. We did not intend to suggest a direct metric-to-metric equivalence, but rather to explore how each performs within a shared diagnostic context.”

"- ROC curves should include 95% CI bands; DeLong comparisons should present ΔAUC with CIs, not only p-values. Please add calibration curves, distribution plots of the ratio by class (ATTR vs non-ATTR), and a Bland–Altman/repeatability plot for voltage measurements if feasible. A transparent model card (TRIPOD adherence table) is recommended. "

Details regarding the ROC curves have been added, including 95% confidence intervals and ΔAUC values with corresponding 95% CI (Table 4, 6-9, 12-13). We apologize, but MedCalc does not support the generation of calibration curves for logistic regression. Based on our verification, the Hosmer–Lemeshow test is the only available method for assessing calibration within the software. The distribution plot of the PWT/minQRS ratio in ATTR-CA patients versus those without ATTR-CA has been added (Figure 2). Unfortunately, it was not feasible to generate a Bland–Altman/repeatability plot for voltage measurements over such a short time interval. TRIPOD adherence table was prepared.

"- The study is retrospective and single-center, yet these limitations are not sufficiently acknowledged or discussed. The patient selection is based on referral for [99mTc]Tc-DPD scintigraphy, which may lead to referral and spectrum bias. The proportion of patients diagnosed with AL amyloidosis (12.6%) and ATTR-CA (22.4%) raises the question of how representative this cohort is of broader clinical practice. A paragraph in the Discussion should explicitly address the issue of generalizability, particularly how this approach would perform in primary cardiology clinics or in asymptomatic carriers of TTR mutations."

The study limitations have been incorporated into the Discussion: “The main limitations of the study are its retrospective and single-center design, which may introduce potential selection bias. Similar to the studies dedicated to the development of the T-amylo and TCAS scores, as well as the posterior wall thickness criterion with a cut-off of ≥14 mm—all of which were also retrospective—we included consecutive patients who underwent both bone scintigraphy and evaluation for clonal dyscrasia. This approach may limit the generalizability of results [6-8]. Selection of patients who underwent bone scintigraphy clearly led to underrepresentation of the AL-CA group, as some patients diagnosed with AL-CA amyloidosis—particularly those in more advanced stages—were directly referred for hematologic treatment without undergoing [99mTc]Tc-DPD scintigraphy. The previous studies on development of TCAS score and the posterior wall thickness criterion with a cut-off of ≥14 mm addressed this issue by excluding all AL-CA patients [6,8]. A different strategy was adopted in the T-amylo study, where five patients with AL-CA were included in the “ATTR-CA excluded” group. However, none of the studies validated their scoring systems in the AL-CA group [6-8]. Our study also lacks proper validation in patients with AL-CA amyloidosis. Although including the AL-CA group made our study more reflective of real-life clinical practice, the relatively small sample size of this subgroup may limit the possibility of making reliable comparisons between study groups. Another limitation is lack of validation of our results in an external, independent prospective cohort.”

Round 2

Reviewer 2 Report

Comments and Suggestions for Authors

The revisions substantially have addressed the reviewer’s core points.